# Spiking neurons from tunable Gaussian heterojunction transistors

Megan E. Beck[1], Ahish Shylendra[2], Vinod K. Sangwan [1], Silu Guo[1], William A. Gaviria Rojas[1], Hocheon Yoo[1], Hadallia Bergeron[1], Katherine Su[1], Amit R. Trivedi[2] & Mark C. Hersam [1,3,4 ✉]

Spiking neural networks exploit spatiotemporal processing, spiking sparsity, and high inter-neuron bandwidth to maximize the energy efficiency of neuromorphic computing. While conventional silicon-based technology can be used in this context, the resulting neuron-synapse circuits require multiple transistors and complicated layouts that limit integration density. Here, we demonstrate unprecedented electrostatic control of dual-gated Gaussian heterojunction transistors for simplified spiking neuron implementation. These devices employ wafer-scale mixed-dimensional van der Waals heterojunctions consisting of chemical vapor deposited monolayer molybdenum disulfide and solution-processed semiconducting single-walled carbon nanotubes to emulate the spike-generating ion channels in biological neurons. Circuits based on these dual-gated Gaussian devices enable a variety of biological spiking responses including phasic spiking, delayed spiking, and tonic bursting. In addition to neuromorphic computing, the tunable Gaussian response has significant implications for a range of other applications including telecommunications, computer vision, and natural language processing.

[1] Department of Materials Science and Engineering, Northwestern University, Evanston, IL 60208, USA. [2] Department of Electrical and Computer Engineering, University of Illinois, Chicago, IL 60607, USA. [3] Department of Chemistry, Northwestern University, Evanston, IL 60208, USA. [4] Department of Electrical and Computer Engineering, Northwestern University, Evanston, IL 60208, USA. ✉email: m-hersam@northwestern.edu

The energy efficiency of spiking neural network (SNN)-based artificial intelligence (AI) can be enhanced by neuromorphic hardware, consisting of spiking neuron-synapse circuits. Since conventional silicon-based complementary metal-oxide-semiconductor (CMOS) transistors do not intrinsically emulate the time-dependent conductance of ion channels in biological neurons, complicated multi-transistor circuits are required for CMOS-based SNNs, thus limiting very-large-scale integration (VLSI) density[1–8]. For example, CMOS-based neuron circuits that achieve multiple spiking modes require at least 20 transistors that must adhere to stringent design constraints in addition to current-based addressing of several branches per neuron[1–3]. Alternatively, IBM TrueNorth[9] and SpiNNaker[10] utilize digital processing of spiking neurons that is seemingly more conducive to VLSI design. However, due to limited chip area, digital cores must multiplex several spiking neurons, which compromises the parallelism of a biological spiking network.

To address the limitations of silicon-based SNN circuits, alternative materials are being explored that allow the encoding of neuromorphic functionality directly at the device level. While memristors[11], memtransistors[12,13], domain-wall memories[14], metal-insulator-transition (MIT) devices[15], multi-gated transistors[16,17], and Gaussian synapses[18] have been developed for scalable implementation of synaptic functions, approaches for realizing spiking neurons are relatively lacking. For example, neuristors based on MIT devices have been reported, but this design suffers from low gain and limited output swing[19,20]. A diffusive memristor coupled with a capacitor has further been shown to exhibit a spiked response, but this demonstration lacks the biophysical characteristics of a neuron spike and runtime neural dynamic adaptation[21]. Leaky integrate and fire spiking neurons have also been achieved by combining a memristor with CMOS transistors[22], but the number of necessary circuit elements remains large. In addition, leaky integrate and fire spiking neurons have been proposed using the magneto-electric effect[23], but this implementation dissipates energy continuously, resulting in poor energy efficiency. A spiking neuron exploiting the abrupt state transition and hysteresis in ferroelectric field-effect transistors has also been shown[24], but this approach is limited to spike frequency adaptation, whereas biological neurons exhibit a variety of other spiking behaviors (e.g., phasic and tonic spiking or bursting)[25]. Ferroelectricity is also highly susceptible to

temperature variations[26], which creates instabilities in ambient operating conditions. Finally, photonic implementations of spiking neurons have recently been discussed based on phase-changing materials[27]. While this strategy is promising for high speed and high bandwidth neural processing, the optical spiking neuron does not exhibit biophysical characteristics.

In contrast, devices fabricated from low-dimensional materials take advantage of weak electrostatic screening to enable gate-tunable electronic properties that hold promise for spiking neurons. In particular, the incorporation of atomically thin semiconducting materials into gate-tunable p-n heterojunctions results in an antiambipolar response with Gaussian transfer curves[28–38]. While this behavior has been used for analog signal processing[37,39], logic devices[30,35,38], and photodetectors[28,32,34], the single-gated geometries used previously do not provide sufficient control over the Gaussian current-voltage characteristic to enable efficient neuromorphic functionality. Here, we report the scalable fabrication of dual-gated Gaussian heterojunction transistors (GHeTs) based on mixed-dimensional van der Waals heterojunctions[40] consisting of monolayer molybdenum disulfide (MoS$_2$) grown via chemical vapor deposition (CVD) and solution-processed semiconducting single-walled carbon nanotubes (CNTs). The dual-gated geometry provides full tunability of the Gaussian transfer curve, thereby enabling simplified circuits that exhibit a variety of neuronal spiking responses including phasic spiking, delayed spiking, and tonic bursting that hold promise for neuromorphic computing and related AI technologies.

## Results

**Device fabrication.** Monolayer MoS$_2$ was specifically selected as the n-type material for our p-n heterojunction because of its atomically thin nature, processing stability, and large-area compatibility via CVD. Solution-processed CNTs were the ideal candidate for the second semiconducting material because of their p-type/ambipolar characteristics, ability to conform over arbitrary nonplanar surfaces, and desired band alignment with MoS$_2$[28,41]. Therefore, a recently reported self-alignment method[37] was adapted to large-area photolithography to enable the fabrication of dual-gated GHeTs from MoS$_2$ and CNTs. As shown in Fig. 1a, the undercut profile in developed negative photoresist combined with directional metal evaporation and

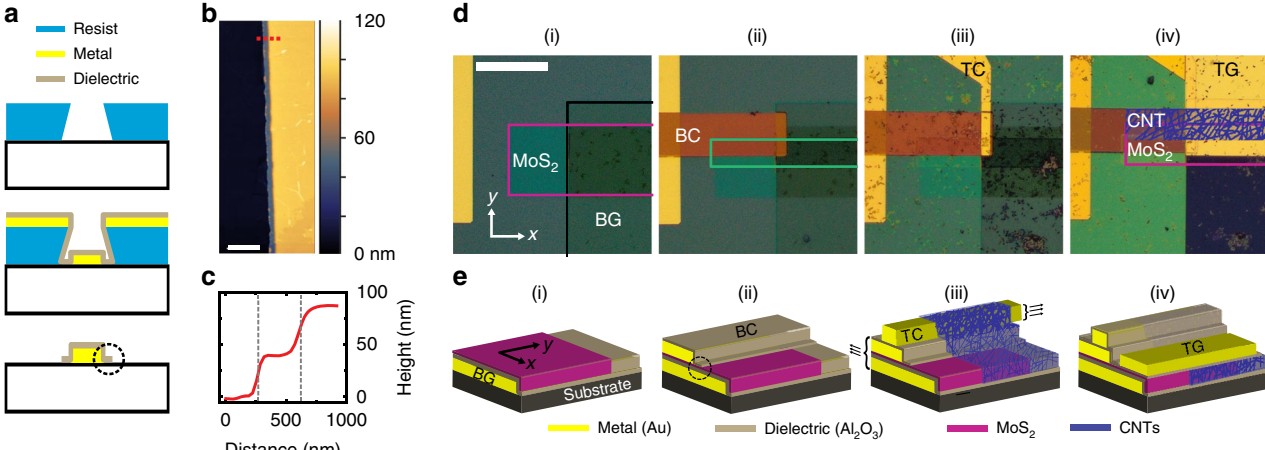

**Fig. 1 Fabrication of Gaussian heterojunction transistors. a** Photolithography-based self-aligned fabrication, which is enabled by resist undercuts that are controlled by spin-coating speeds, exposure time, and development time. **b** Atomic force microscopy topography image of an electrode and dielectric extension (2 μm scale bar) corresponding to the dashed circle in **a**. **c** Height profile corresponding to the dashed red line in **b**, revealing a sub-exposure-wavelength extension of ~300 nm on monolayer MoS$_2$. **d** Optical micrographs of the fabrication process (50 μm scale bar). **e** Three-dimensional rendering of the device structure throughout fabrication.

conformal atomic layer deposition (ALD) of a dielectric oxide results in an encapsulated metal electrode with a self-aligned dielectric extension. Atomic force microscopy (AFM) is used to evaluate both the quality and length of the dielectric extension as shown in Fig. 1b. The height profile in Fig. 1c corresponding to the dashed line in Fig. 1b shows a sub-exposure-wavelength dielectric extension of ~300 nm. The steps in the height profile correspond to the ALD $Al_2O_3$ (~35 nm) and metal electrode (~50 nm).

Optical micrographs of the fabrication process are shown in Fig. 1d. (i) CVD-grown monolayer $MoS_2$[42] is transferred onto a self-aligned local bottom gate (BG, outlined in black) and patterned (outlined in pink) using reactive ion etching (RIE). (ii) A self-aligned bottom contact (BC) is then fabricated on $MoS_2$ followed by an additional patterning and growth of a thin $Al_2O_3$ dielectric (outlined in green, distinct from the self-aligned $Al_2O_3$ dielectric) on part of the $MoS_2$ to act as an etch mask for subsequent RIE processing. (iii) The metal top contacts (TC) are then deposited directly on top of the BC followed by transfer of a network of semiconducting CNTs[43,44] over the entire substrate, after which RIE is used to define the CNT network (purple lines) with deterministic overlap of the $MoS_2$ region. (iv) Finally, an $Al_2O_3$ dielectric is grown via ALD over the entire substrate, and local top gates are patterned over the junction region. Three-dimensional renderings of the fabrication process shown in Fig. 1e correspond to: (i) $MoS_2$ is transferred onto the bottom gate (BG) and etched; (ii) Bottom contact (BC) is deposited on $MoS_2$ (dielectric extension in dashed circle); (iii) Top contact (TC) is deposited on the BC followed by semiconducting single-walled carbon nanotube (CNT) network transfer and etching; (iv) ALD $Al_2O_3$ is used to cover the entire device structure, after which the top gate (TG) is deposited and patterned. Note, the ALD etch mask outlined in green in Fig. 1d (ii) is intentionally not shown in Fig. 1e to better illustrate the self-aligned and semi-vertical device architecture but is shown in Supplementary Fig. 1. In this dual-gated semi-vertical device, the multiple current paths through the semiconducting materials increase the versatility of device operation (see Supplementary Fig. 2).

**Electrical characterization.** Dual-gated control transistors from the constituent semiconductors were characterized to confirm the desired individual material properties. Transfer and output measurements of the 50 μm $MoS_2$ and CNT dual-gated devices are shown in Supplementary Fig. 3 and Supplementary Fig. 4. The $MoS_2$ devices exhibit n-type behavior while the CNT devices exhibit ambipolar behavior. Both materials show dual-gate tunability of threshold voltages, indicating that the heterojunction should also exhibit dual-gate-tunable diode properties and the desired antiambipolar response.

The GHeTs were first characterized by biasing the bottom and top gates independently with the source voltage ($V_S$) grounded. Figure 2a shows selected output curves corresponding to different top gate voltages ($V_{TG}$) for $V_{BG}$ = 0 V. The top gate modulates the output response of the GHeT from a rectifying diode at $V_{TG}$ = −6 V (orange) to an inverted polarity rectifying diode at $V_{TG}$ = 6 V (purple) due to band-to-band tunneling between the $MoS_2$ and CNTs as has been previously reported[37,41,45]. Supplementary Fig. 5 shows additional sets of output curves corresponding to $V_{BG}$ = 6 V and −6 V. To further characterize the GHeT rectifying behavior, Fig. 2b shows the rectification ratio (defined here as $I_D$ at $V_D$ = 1 V divided by $I_D$ at $V_D$ = −1 V) extracted from the corresponding transfer curves. The top and bottom gates are both able to modulate the diode rectification ratio. For $V_{TG}$ > 0 V, the rectification ratio can be tuned by over two orders of magnitude, including reversal of the rectification direction (i.e., rectification

ratio < 1) for $V_{TG}$ > 2 V. For $V_{TG}$ < 0, modulation by the bottom gate is more evident with tunability of the rectification ratio by over two orders of magnitude at $V_{TG}$ = −6 V. The rectification ratios for $V_{BG}$ from 6 V to −6 V with 1 V increments are shown in Supplementary Fig. 6. The GHeT transfer curve as a function of $V_{TG}$ for independent biasing where $V_{BG}$ is held constant throughout the measurement is shown in Fig. 2c. As the $V_{BG}$ setpoint is varied, the peak position of the antiambipolar response can be tuned from $V_{TG}$ = 2 V to $V_{TG}$ = −3 V (see Supplementary Fig. 7 for additional curves). Note that the behavior to the right of the peak is correlated to electrostatic control of the CNT film while the behavior to the left of the peak is correlated to electrostatic control of the $MoS_2$. Thus, the top gate can fully modulate the CNTs at all $V_{BG}$, as evidenced by a distinct OFF state at $V_{TG}$ ~4 V in Fig. 2c and the negative transconductance ($g_m$) from the peak voltage to $V_{TG}$ = 4 V shown in Supplementary Fig. 8. Due to low dielectric screening by CNT networks[45], the top gate can partially modulate the n-type $MoS_2$ as evidenced by positive $g_m$ in Supplementary Fig. 8 for $V_{BG}$ < 0 V and negative $V_{TG}$. For $V_{BG}$ > 0, the negative bias field from the top gate through the CNT network is not sufficiently strong to fully deplete the $MoS_2$ that has been driven into accumulation by the bottom gate, resulting in loss of dual-gate control for the left side of the antiambipolar response. The corresponding GHeT transfer curve and plot of $g_m$ as a function of $V_{BG}$ (see Supplementary Fig. 9) indicate that the bottom gate can fully modulate $MoS_2$ for all biases but is unable to fully modulate the CNT network completely at any bias due to stronger dielectric screening of the bottom gate bias by the continuous $MoS_2$ monolayer. The loss of dual-gate control for independent biasing indicates that the current flows primarily through the overlap region of the GHeT (see Supplementary Fig. 2).

Alternatively, the GHeT can be operated in a dependent biasing scheme to combine the modulation of the CNTs by the top gate and the modulation of the $MoS_2$ by the bottom gate, resulting in enhanced electrostatic control of the device response. Since the $Al_2O_3$ dielectric layer for both gates is ~35 nm thick, the fields from the top gate and the bottom gate are equivalent for the same bias. Figure 3a shows the transfer response of the GHeT when $V_{BG}$ and $V_{TG}$ are changed together throughout the measurement with a constant offset ranging from −3 V to 3 V while $V_S$ is grounded. Rectification ratios extracted from the corresponding transfer curves for dependent gate operation can be tuned by over three orders of magnitude as shown in Supplementary Fig. 10. The Gaussian fits of the antiambipolar response (see Supplementary Fig. 11) illustrate that changing the offset between the gates from −3 V to 3 V can shift the peak position from $V_{TG}$ = −3 V to 0.5 V without a loss of symmetry in the antiambipolar response and without a substantial loss in the peak current. As shown in Supplementary Fig. 12 for $V_{TG}$ − $V_{BG}$ = 0 V, 85% of working devices fabricated over an 0.5 × 0.5 cm area exhibit a Gaussian transfer response. The average peak position of these 14 devices was −0.42 V ± 0.55 V, and the average full-width-half-maximum (FWHM) was 2.92 V ± 0.48 V.

Combining $V_D$ modulation with dual-gate tunability results in further control over the peak height, position, and FWHM of the GHeT antiambipolar response. Figure 3b shows that for dependent operation of the gates at $V_{TG}$ − $V_{BG}$ = 0 V, variable $V_D$ modulates the peak height while maintaining the peak position with minimal change in FWHM. On the other hand, by changing the dependent gating offset $V_{TG}$ − $V_{BG}$ from 3 V to −3 V and $V_D$ from 0.8 V to 0.4 V, the peak position can be tuned while maintaining the peak height and FWHM as shown in Fig. 3c. Finally, Fig. 3d shows that the peak height and peak position can be maintained while the FWHM is varied by switching between dependent and independent gate operation. Independent

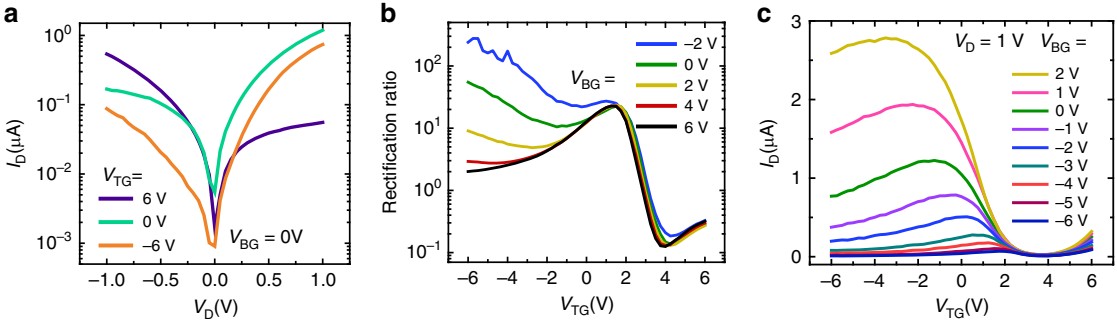

**Fig. 2 Independent gate operation of Gaussian heterojunctions transistors. a** Representative $I_D$-$V_D$ curves for $V_{BG} = 0$ V, illustrating the $V_{TG}$ tunability of the rectifying diode response, including inverted polarity of the rectification direction (orange versus purple). **b** Rectification ratios ($I_D$ at $V_D = 1$ V divided by $I_D$ at $V_D = -1$ V) of the diode for various biasing conditions, showing tunability by both $V_{TG}$ and $V_{BG}$. **c** $I_D$-$V_{TG}$ for different values of $V_{BG}$, exhibiting tunability of the peak position of the antiambipolar response. All measurements were performed in ambient at room temperature with $V_S = 0$ V.

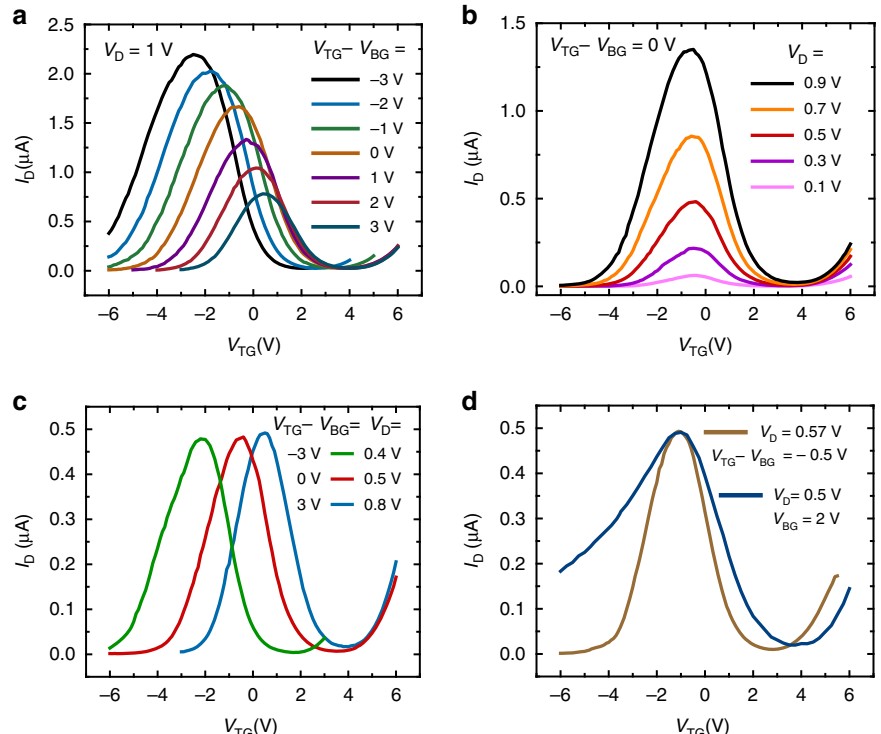

**Fig. 3 Dependent gate operation of Gaussian heterojunction transistors. a** $I_D$-$V_{TG}$ for simultaneous sweeping of $V_{BG}$ with $V_{TG}$ with controlled offsets, showing control of both sides of the antiambipolar response and of the peak position. **b** $I_D$-$V_{TG}$ for varied $V_D$, showing that the height of the Gaussian transfer response can be controlled while maintaining the peak position. **c** $I_D$-$V_{TG}$ for changing $V_D$ and $V_{TG}$-$V_{BG}$, indicating that the peak position can be controlled while maintaining the height of the Gaussian transfer response. **d** $I_D$-$V_{TG}$ comparing dependent and independent gate biasing with an adjusted $V_D$, showing modulation of the FWHM of the Gaussian transfer response while maintaining the height and peak position. All measurements were performed in ambient at room temperature with $V_S = 0$ V.

modulation of peak height, position, and FWHM confirms that the mixed-dimensional MoS$_2$-CNT GHeT possesses a fully tunable Gaussian transfer response in a single heterojunction device.

**Spiking neuron demonstration.** The ability to tune the antiambipolar response of the GHeT enables a variety of applications including Hodgkin-Huxley (HH) spiking neurons[46]. A circuit-level representation of the HH model for biological neurons is shown in Fig. 4a where Na$^+$ ions injected into the neuron lead to spike generation in the membrane potential, $V_m$, while released K$^+$ ions reset $V_m$. Capacitance, $C_m$, and leakage conductance, g$_L$, represent the bilayer of the neuron membrane. The conductance

of the Na$^+$ (K$^+$) ions channels is modeled by g$_{Na}$ (g$_K$). Figure 4b, c show the temporal evolution of g$_{Na}$ and g$_K$ as described by the HH model. The activation of the K$^+$ ion channel is delayed and g$_K$ increases with increasing $V_m$. The time-dependent evolution of g$_K$ can be represented by the delayed turn-on of an n-channel metal-oxide semiconductor (NMOS) transistor where a voltage, $V_m$, is applied to the gate through a resistive-capacitive load. Meanwhile, the behavior of the Na$^+$ ion channel is more complex and requires a peaked time-dependent response, where the peak conductance increases but the delay to reach the peak conductance is reduced with increasing $V_m$. To efficiently capture the more complex transient behavior of g$_{Na}$, the dual-gated GHeT antiambipolar response is exploited as described in Supplementary Fig. 13.

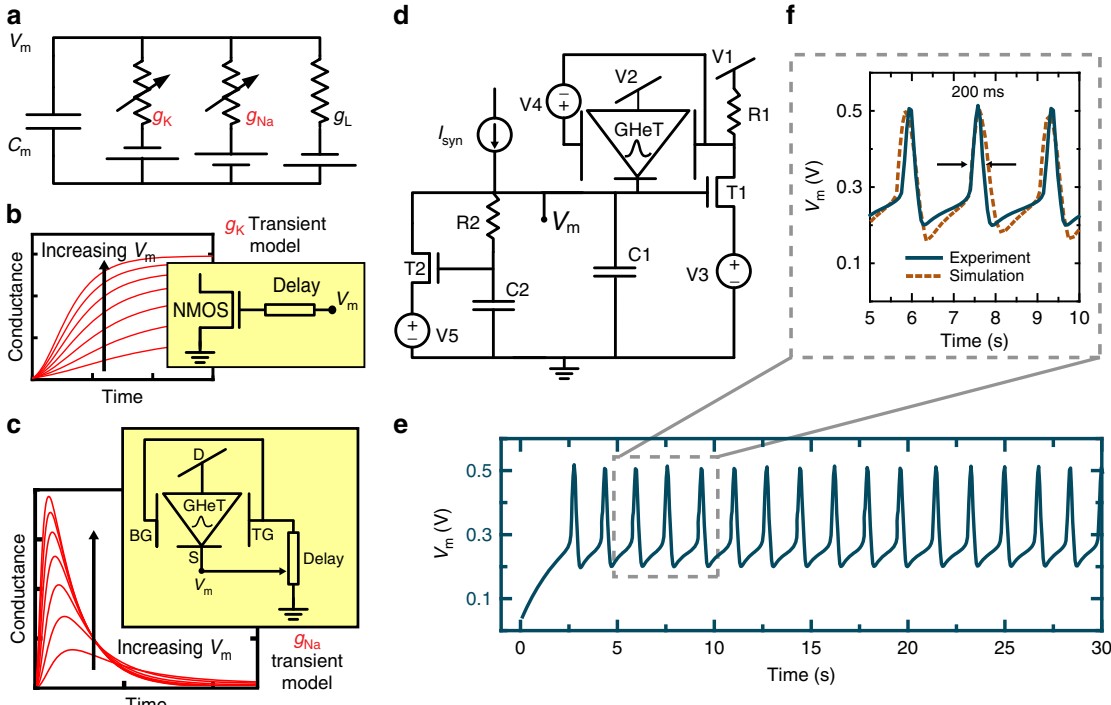

**Fig. 4 Hodgkin-Huxley spiking neuron using a Gaussian heterojunction transistor. a** Circuit-level equivalent of the Hodgkin-Huxley model. **b** Temporal evolution of $g_K$, which can be represented by the delayed turn-on of an NMOS transistor. **c** The more complex transient behavior of $g_{Na}$, which can be mimicked using the antiambipolar characteristics of the GHeTs. **d** Full circuit diagram for the experimental spiking neuron. **e** Experimental results for the first 30 sec based on the GHeT neuron circuit detailed in **d**. **f** From dashed region in **e** the neuron spike FWHM is measured to be ~200 ms. Simulations (dashed brown) of the GHeT neuron circuit agree well with the experimental results. All measurements were performed in ambient conditions at room temperature.

Figure 4d details the full circuit used for the experimental spiking neuron demonstration based on a single $MoS_2$-CNT GHeT device, n-type field-effect transistors (T1 and T2), and a few passive elements (R1, R2, C1, and C2). Voltage sources, V3 and V5, were connected at the source electrodes of T1 and T2 to allow threshold voltage programmability for the field-effect transistors. The GHeT and circuit components T1-R1-C1 emulate $g_{Na}$, while circuit components T2-R2-C2 emulate $g_K$ (see Supplementary Fig. 14). Before application of a synaptic current, the GHeT is in an OFF state due to a large positive gate bias (~V1), corresponding to position 1 in Supplementary Fig. 15 and Supplementary Table 1. For sufficiently high $I_{syn}$, C1 and C2 integrate $I_{syn}$ and the OFF current of the GHeT ($I_{OFF}$). In other words, the voltage at the GHeT source, $V_m$, increases with time proportional to $I_{syn} + I_{OFF}$ (position 2 in Supplementary Fig. 15). As $V_m$ exceeds the threshold voltage of T1, the voltage applied to the gates drops quickly from V1 to near 0 V, resulting in a negative relative gate voltage, $V_{TG} - V_m$. This condition drives the GHeT from its OFF state to the peak ON state, going through the region of negative transconductance. The increased current, $I_{PEAK}$, causes a sharp increase in the slope of $V_m$ in proportion to $I_{syn} + I_{PEAK}$. As $V_m$ continues to increase, $V_{TG} - V_m$ continues to decrease, thereby accessing the left side of the Gaussian response and resulting in a decreasing current and positive $g_m$ (position 3 in Supplementary Fig. 15). At this point, $V_m$ has reached the threshold voltage of T2, and the delayed $g_K$ channel is able to dominate and reset $V_m$ below the threshold voltage of T1 (position 4 in Supplementary Fig. 15). This spiking and resetting behavior, experimentally shown in Fig. 4e, will continue as long as $I_{syn} + I_{OFF}$ is sufficiently high.

Figure 4f shows that simulations performed on the Cadence Virtuoso platform using the Spectre simulator for a prototypical

GHeT (brown dashed) agree with the experimental spiking response of the circuit (blue) with a temporal FWHM that is ~200 ms. The energy consumption of this GHeT-based spiking neuron circuit is ~250 nJ per spike, which can be reduced by orders of magnitude by decreasing the channel width, circuit capacitances (C1 and C2), and gate dielectric thickness of the GHeT as well as by custom design and on-chip integration of T1 and T2 transistors. These modifications will also substantially increase the operating speed of the spiking neuron circuit. The measured gate voltage and current from the GHeT during the full 30 sec of constant spiking are shown in Supplementary Fig. 16, thereby confirming that the GHeT is responsible for the spiking behavior. In addition to capacitance values and $I_{syn}$ affecting the spiking response (Supplementary Fig. 17), experimental results in Supplementary Fig. 18 and simulation results in Supplementary Fig. 19 show that the offset between the gates can be used to further control the spiking response. In particular, constant spiking only occurs when the circuit operating region (4 V to −1 V) contains both negative and positive $g_m$ values near $I_{PEAK}$ of the Gaussian transfer response, demonstrating that the anti-ambipolar response is required to correctly mimic $g_{Na}$ in the HH model of a spiking neuron.

Additional simulations show that multiple biological spiking neuron responses can be achieved with GHeT-based circuits by modifications to how the GHeT is biased by the top and bottom gates. Simulations using the experimental circuit (see Supplementary Fig. 20), where the GHeT experiences dependent biasing, show that constant spiking occurs for a constant $I_{syn}$ of 40 nA (Fig. 5a), whereas, if $I_{syn}$ increases linearly from 0 to 80 nA, then the spiking frequency increases (Fig. 5b). A slight modification to the GHeT-based circuit that allows for independent gate biasing (see Supplementary Fig. 21) enables additional functionality such

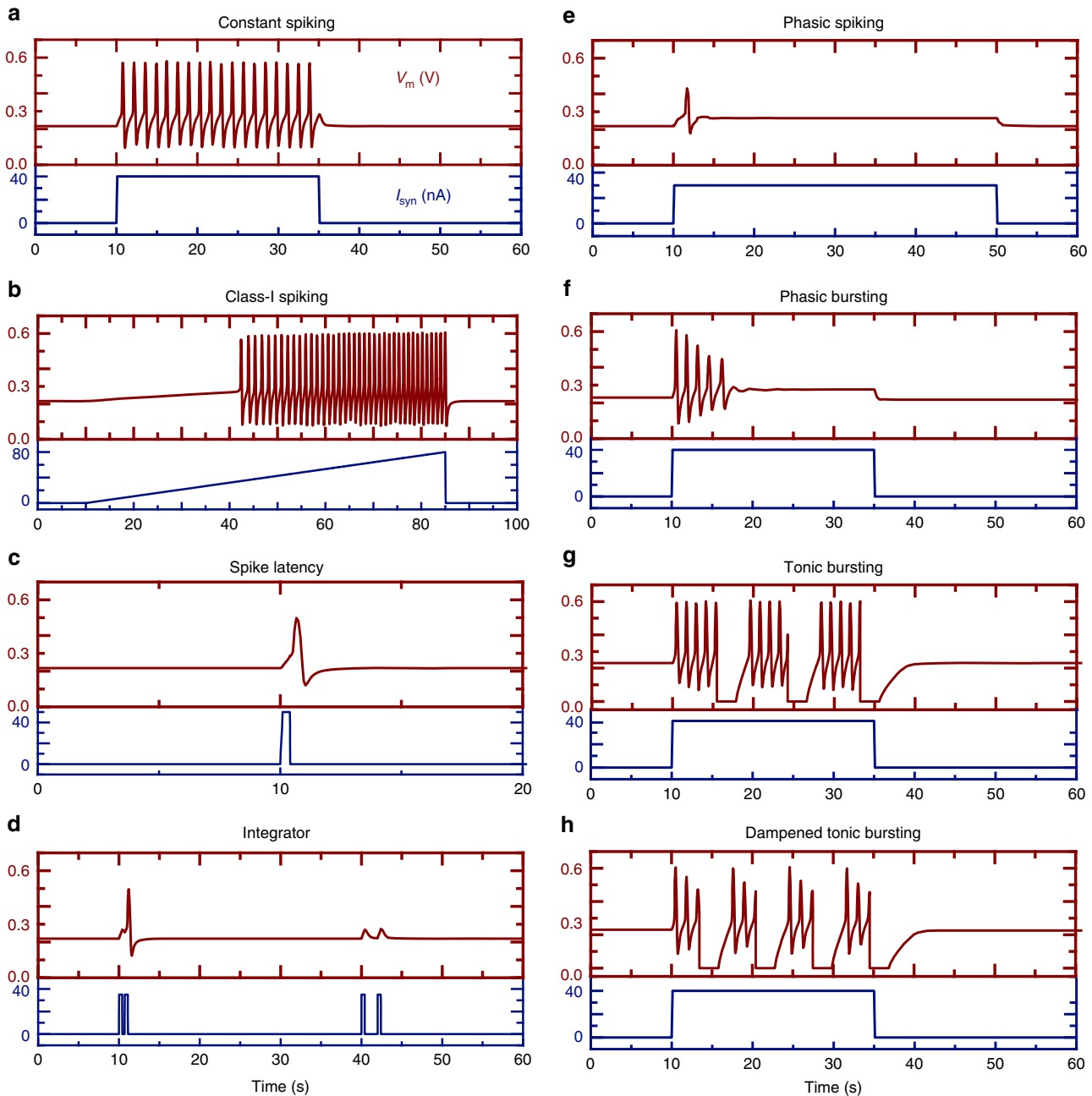

**Fig. 5 Simulated spiking responses from GHeT-based circuits. a, b** For the experimentally demonstrated circuit (see Supplementary Fig. 20), **a** constant spiking occurs when a constant $I_{syn}$ is applied and **b** spiking frequency increases with increasing $I_{syn}$ to mimic Class-I spiking. **c–e** Using independent gate operation (see circuit in Supplementary Fig. 21), **c** spike latency, **d** integrator, and **e** phasic spiking occur for various $I_{syn}$. **f–h** Additional minor modifications to the GHeT neuron circuit enable **f** phasic bursting (see circuit in Supplementary Fig. 22), **g** tonic bursting (see circuit in Supplementary Fig. 23), and **h** dampened tonic bursting (see circuit in Supplementary Fig. 24).

as spike latency, integrator, and phasic spiking responses. For example, Fig. 5c illustrates spike latency where a single neuron spike occurs after a 0.4 sec $I_{syn}$ pulse of 50 nA. The integrator response in Fig. 5d shows that a neuron spike occurs when the 0.4 sec $I_{syn}$ pulses of 35 nA are within 0.3 sec of each other (first set of pulses) but does not occur when the pulses are 1.6 sec apart (second set of pulses). In this circuit, a constant $I_{syn}$ of 40 nA results in a single neuron spike, mimicking phasic spiking (Fig. 5e). Further modification of the circuit to connect the top and bottom gates through an inverter, diode, resistor, and capacitor (see Supplementary Fig. 22) results in phasic bursting for a constant $I_{syn}$ of 40 nA (Fig. 5f). Incorporating a Schmitt

Trigger and two additional transistors (see Supplementary Fig. 23 and Supplementary Fig. 24) results in tonic bursting (Fig. 5g) and dampened tonic bursting (Fig. 5h) for a constant $I_{syn}$ of 40 nA. With incremental modifications to the original circuit, these simulations highlight the versatility of GHeT-enabled circuits to have runtime programmability of spiking threshold and spiking modes that are desirable for spiking neuron applications.

## Discussion
By creating a device with intrinsic neuronal responses, it is possible to significantly simplify spiking neuron implementations. In

particular, the use of mixed-dimensional MoS$_2$-CNT van der Waals heterostructures and a semi-vertical, dual-gated geometry results in a smaller device footprint with superior electrostatic control compared to other antiambipolar demonstrations. Not only is the fundamental behavior of the Na$^+$ ion channel of a biological neuron captured by the GHeT in a simple circuit, but by exploiting the dual-gated programmability both through independent and dependent biasing, it is possible to achieve eight different biological neuron responses, five of which are achieved using a single GHeT, two transistors, two capacitors, and two resistors. Additionally, the fabrication process for GHeT-based spiking neurons is compatible with previous demonstrations of monolayer MoS$_2$ memtransistor-based synapses[12,13], enabling scalable implementations of biomimetic neuromorphic platforms.

More broadly, since CMOS transistors cannot natively mimic the Gaussian response demonstrated here, CMOS-based digital designs implement Gaussian functions with complex circuits and look-up tables while analog CMOS circuits suffer from limited programmability and high bias current[47]. Thus, the tunable GHeT Gaussian antiambipolar response is applicable to hardware-level implementations of spiking neurons as well as other artificial learning paradigms. For example, several natural language processing algorithms require Gaussian functions to build statistical distributions of speech and phoneme characteristics[48]. Similarly, neural networks used in machine learning often account for uncertainties in Bayesian inference using weight densities represented by a mixture of Gaussian functions[49]. GHeTs are also likely to be useful for highly efficient computer vision algorithms in artificial neural networks that rely on the tunability and intrinsic filtering ability of a Gaussian response[50]. Given that the complexity of CMOS-based implementations is a bottleneck for many learning models, the simplification of the Gaussian response to a single GHeT circuit element is expected to accelerate the realization of AI-based technologies.

## Methods

**Fabrication of Gaussian heterojunction transistors**. All photolithography steps were performed on a Suss MABA6 Mask Aligner with an exposure wavelength of 365 nm and an exposure intensity of 9 mW cm$^{-2}$ using resist developer RD6 (Futurrex, Inc.) and liftoff for 1 h in Remover PG (MicroChem) at 70 °C unless specified otherwise. The devices were fabricated on undoped Si/300 nm SiO$_2$ substrates. Following the self-aligned process described in Fig. 1a using negative resist (NR9-1000PY, Futurrex), the bottom gate metal of 10 nm Cr/10 nm Au/4 nm Al was thermally evaporated (Kurt J. Lesker, Nano 38) followed by atomic layer deposition (ALD, Cambridge Nanotech ALD S100) of ~35 nm of Al$_2$O$_3$ grown at 100 °C. The 4 nm of Al oxidizes readily in ambient conditions and acts as a seeding layer for the growth of the ALD dielectric on the Au metal surface. A monolayer of MoS$_2$ grown on a sapphire substrate using solid-precursor CVD was then transferred onto the local bottom gate structure using a wet polycarbonate-assisted transfer process. The MoS$_2$ monolayer was patterned using a positive resist bilayer of polymethylglutarimide (PMGI, MicroChem) and S1813 (MicroChem), and etched by reactive ion etching (RIE, Samco RIE-10NR) using 50 sccm Ar at 13.3 Pa and 50 W for 20 sec. The PMGI/S1813 bilayer was used to minimize S1813 resist residue on the remaining MoS$_2$ monolayer but required overnight liftoff. After repeating the self-aligned process, the encapsulated bottom contacts (4 nm Ti/40 nm Au/4 nm Al, 35 nm Al$_2$O$_3$) were patterned and deposited on the etched MoS$_2$ monolayer. The portion of the final MoS$_2$ film that is designed not to be covered by the film of semiconducting single-walled carbon nanotubes (CNTs) was protected from further etching by a patterned region of ~5 nm Al$_2$O$_3$. The top contacts (10 nm Cr/70 nm Au) were deposited on top of the encapsulated bottom contacts in preparation for the CNT film. The optimized concentration (~10 tubes per μm transferred) of solution-processed P2 single-walled semiconducting CNTs with 99% semiconducting purity obtained via density gradient ultracentrifugation was vacuum filtered onto a cellulose membrane (VMWP, 0.05 μm pore size, Millipore Sigma) and acetone-bath transferred overnight on the entire substrate. The film of CNT was patterned using S1813 and etched by RIE using 20 sccm O$_2$ at 26.5 Pa and 100 W for 15 sec. The substrate was rinsed briefly (<30 sec) with acetone to remove all but a few nanometers of the residual S1813 film. The S1813 residue acts as an encapsulant to minimize doping of the CNTs from the top gate dielectric ALD of ~35 nm Al$_2$O$_3$ that was deposited over the entire substrate. The top gate metal (10 nm Cr/60 nm Au) was then patterned and deposited to overlap the entire device region.

**Materials characterization and electrical measurements**. The thicknesses of the different device layers were characterized by atomic force microscopy (AFM) in ambient using an Asylum Cypher AFM. All electrical measurements were performed in ambient on a Cascade MicroTech semi-automated probe system using a Keithley 4200 semiconductor analyzer.

**Statistics**. Devices were fabricated over an area of 0.5 × 0.5 cm with 85% yield. The $I_D$–$V_{TG}$ antiambipolar response for 14 distinct devices is shown in Supplementary Fig. 12b. Values for the histograms in Supplementary Fig. 12c were obtained by fitting the raw data to

$$y = y_0 + Ae^{-0.5\left(\frac{x-x_c}{w}\right)^2} \tag{1}$$

where $x_c$ is the peak position and $w$ is the FWHM. The average peak position was -0.42 V ± 0.55 V, and the average FWHM was 2.92 V ± 0.48 V. Source data underlying Supplementary Fig. 12c are provided as a Source Data file. Note, the data presented in Figs 2 and 3 correspond to the same device.

**Spiking neuron demonstration**. The experimental demonstration of a constant spiking neuron was achieved from the circuit in Fig. 4d using V1 = 4 V, V2 = 1 V, V3 = −230 mV, V4 = 3 V, V5 = −280 mV, R1 = 100 kΩ, R2 = 1 MΩ, C1 = 440 nF, C2 = 220 nF, $I_{syn}$ = 1 nA and commercial field-effect transistors, BSR802N L6327 (Mouser). Circuits simulations were performed using the Cadence Virtuoso platform using Spectre simulator with a look-up table-based Verilog-A model developed for a prototypical MoS$_2$-CNT GHeT as well as for the commercial transistors based on experimental charge transport characteristics. Other passive and active elements were obtained from the Analog library available within Virtuoso. Supplementary Table 2 contains the parameters used in the simulations of the GHeT-based circuits used in Figs 4 and 5.

## Data availability

The data that supports the findings of this study are available within the paper and its supplementary files or available from the corresponding author upon request. The source data underlying Supplementary Fig. 12c are provided as a Source Data file.

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

## Acknowledgements

This research was supported by the 2-DARE program (NSF EFRI-1433510) and the Materials Research Science and Engineering Center (MRSEC) of Northwestern University (NSF DMR-1720139). CVD growth of $MoS_2$ was supported by the National Institute of Standards and Technology (NIST CHiMaD 70NANB14H012). Charge transport instrumentation was funded by an ONR DURIP grant (ONR N00014-16-1-3179). H.B acknowledges support from the NSERC Postgraduate Scholarship-Doctoral Program. M.E.B., W.A.G.R., and H.B. acknowledge support from the National Science Foundation Graduate Research Fellowship Program. This work utilized the Northwestern University Micro/Nano Fabrication Facility (NUFAB), which is partially supported by Soft and Hybrid Nanotechnology Experimental (SHyNE) Resource (NSF ECCS-1542205), the Materials Research Science and Engineering Center (DMR-1720139), the State of Illinois, and Northwestern University.

## Author contributions

M.E.B. and M.C.H. conceived the device structure and designed all the experiments. A.T., M.E.B., and V.K.S. conceived the circuit for the spiking neuron. A.S. and A.T. performed circuit simulations. V.K.S., M.E.B., and W.A.G.R. designed the photomask. M.E.B. optimized the fabrication process, measured and analyzed all the device data, and implemented the circuit experimentally. S.G. assisted with device fabrication. W.A.G.R. assisted with CNT processing. W.A.G.R. and H.Y. contributed to the experimental circuit demonstration. H.B. and K.S. conducted growth and transfer of $MoS_2$. All authors wrote the manuscript and discussed the results at all stages.

## Competing interests

The authors declare no competing interests.
