## [Peer Review File · Nature Communications]

Reviewers' comments:

Reviewer #1 (Remarks to the Author):

The manuscript by Megan E. Beck et al. reports a MoS₂/CNT Gaussian heterojunction transistor which shows an antiambipolar behaviour, which can be integrated into a circuit to enable the integrate-and-fire. This work demonstrates the potential application of Gaussian transistor as a spiking neuron. However, there are major weaknesses in the results and data analysis that need to be addressed.

1. Figure 1d and 1e, step (ii) and (iii) are inconsistent. In figure 1d, if the Al₂O₃ (outlined in green) is deposited there, the TC and BC will be shorted to each other. However, in the 3D schematic, the TC and BC are actually aligned horizontally. Can the author explain why the optical image shows that the TC and BC are deposited orthogonally?

2. Fig. 2a shows that rectifying diode has two polarities at VTG=6V and -6V. The authors attribute this behaviour to a possible band-to-band tunnelling. The two polarities behaviour was reported previously (Small 15, 1804661, 2019) by changing the back gate voltage. This is because MoS₂ and CNT work in different regime with different back gate voltage. I suggest the author to measure the transfer curve of MoS₂ and CNT, and the output curve of MoS₂ and CNT at different gate voltage, respectively. Furthermore, can the authors provide a detailed explanation based on the new results requested above?

3. Fig. 2 and Fig. 3 shows that both top/back gates could modulate the heterojunction. So these two figures actually show the same information. I suggest the authors to combine Fig. 2 and Fig. 3 together and replace Fig. 3 with the dynamic response of the device with pulse measurement that is typically used to characterize the artificial neuron.

4. In page 5, the authors state "The top gate can fully modulate the CNTs at all VBG, as evidenced by the negative transconductance (gm) in Fig. 2c. Due to low dielectric screening by CNT networks, the top gate can also modulate the n-type MoS₂ for VBG < 0 V, as evidenced by positive gm." According to my understanding, it means the top gate can modulate the CNT as the drain current when VTG<0 varies at all VBG. The top gate could also modulate the MoS₂ as the drain current when VTG >0 also varies when VBG <0. By claiming that the drain current shows dependence on gate voltage is clear enough, why do the authors mention gm without showing the gm curve? gm is typically used to show how good the gate could control the channel. Clearly, the authors just want to show that top gate could control both CNT and MoS₂. I suggest to rewrite these sentences.

5. In page 8, the authors state "This condition drives the GHeT through its peak current, IPEAK, causing a sharp increase in the slope of Vm in proportion to Isyn + IPEAK. As Vm continues to increase, VTG - Vm continues to decrease, accessing the positive gm of the GHeT and allowing the delayed gK channel to dominate and reset Vm. As Vm continues to increase, VTG - Vm continues to decrease, accessing the positive gm of the GHeT and allowing the delayed gK channel to dominate and reset Vm." This sentence is very confusing and needs to be revised for better clarity.

Since the device already passed through its peak current, as Vm continues to increase, the VTG - Vm continues to decrease, so the gm should go to the negative regime. Why the authors think it will access the positive gm? Moreover, as Vm continues to increase, the current will drop to 0 according to the I-V curve. Does the C2 start to charge T2 when the current drop to 0? What are the considerations when choosing C1 and C2?

Below are some minor comments/questions about the manuscript:

1. There is a typo in the introduction. It should be "much" instead of "mush".
2. The authors claim wafer-scale production and a yield of 85%. Can the authors show the optical image of the device array fabricated over a 0.5 cm by 0.5 cm area?
3. Figure 4c is very confusing. Please label source, drain, TG and BG of the GHeT clearly and explain the conductance vs time curve for gk.

Overall, this manuscript demonstrated the integrate-and-fire of an artificial neuron by integrating the Gaussian transistor into a circuit. However, the writing is very hard to read and the data analysis is unclear. Considering the broad readership of this journal with different background, I strongly suggest that the authors improve the manuscript to an acceptable level.

Reviewer #2 (Remarks to the Author):

The authors present a new dual-gated device called Gaussian Heterojunction Transistor that shows very good tunability control, which allows (among possibly many other applications) its versatile use for building spiking neurons. I am not familiar with the fabrication processes nor device characteristics, so I cannot comment on that and compare with respect to alternative state-of-the-art solutions. From a system-level point of view of neuromorphic systems and applications, the solution proposed is certainly interesting. However, the authors should emphasize that more devices are required to build a spiking neuron, not just the new heterojunction transistor. For example, in Figs. 14-18 many other transistors plus passive elements (resistors and capacitors) are required.

The authors should compare their work to other alternatives. For example, <https://www.nature.com/articles/s41928-018-0023-2> shows a single device that can emulate a neuron just by adding a single capacitor or resistor. Authors should highlight their advantages with respect to this work.

For fully CMOS solutions, they may compare against, for example, <https://ieeexplore.ieee.org/abstract/document/5537564>, or <https://ieeexplore.ieee.org/abstract/document/5372050>. These use a large number of transistors. But pioneering work of Cullurciello provided a very simple integrate-and-fire with a very reduced number of transistors, like <https://ieeexplore.ieee.org/abstract/document/1175509>.

Reviewer #3 (Remarks to the Author):

In this manuscript (entitled "Spiking Neurons from Tunable Gaussian Heterojunction Transistors"), the authors proposed spiking neurons based on dual-gated tunable Gaussian heterojunction transistors with MoS₂ and CNTs. Hodgkin-Huxley spiking neuron was realized using such a Gaussian heterojunction transistor. With simulations, various biological spiking responses including phasic spiking, delayed spiking, and tonic bursting could be obtained. In general, this work is interesting and suitable for Nature Communications. There are some comments to the authors:

(1) What is the energy consumption per spike of the device? Please compare it with the energy consumption with the biological neurons and artificial neurons developed by other types of devices, metal-insulator-transition (MIT) devices, for example.

(2) Please explain why MoS₂ and CNTs are selected? And compare it with other materials such as oxide semiconductors.

(3) As the heterojunction transistors can be tuned by VTG and VBG, can the properties of the spiking neurons, such as spiking frequency, based on the heterojunction transistors be modulated by VTG and VBG? If yes, please show the modulated properties.

Reviewer #1 (Remarks to the Author):

The manuscript by Megan E. Beck et al. reports a MoS₂/CNT Gaussian heterojunction transistor which shows an antiambipolar behaviour, which can be integrated into a circuit to enable the integrate-and-fire. This work demonstrates the potential application of Gaussian transistor as a spiking neuron. However, there are major weaknesses in the results and data analysis that need to be addressed.

1. Figure 1d and 1e, step (ii) and (iii) are inconsistent. In figure 1d, if the Al₂O₃ (outlined in green) is deposited there, the TC and BC will be shorted to each other. However, in the 3D schematic, the TC and BC are actually aligned horizontally. Can the author explain why the optical image shows that the TC and BC are deposited orthogonally?

Response:

We apologize for the confusion between Fig. 1d and Fig. 1e. The original purpose of these figures was to show different aspects of the device structure. In Fig. 1d (ii), the self-aligned atomic layer deposition (ALD) alumina that prevents shorting between TC and BC cannot be distinguished from BC in the optical image due to the scale. However, it is distinct from the additional region of ALD that is outlined in green in the optical image. The purpose of the green region of ALD is to prevent etching of the MoS₂ when the CNT film is patterned. On the other hand, the aim of Fig. 1e (ii) is to reveal the self-aligned dielectric extending from BC onto the MoS₂ (highlighted by the dashed circle). Thus, the green ALD region is not shown in the 3D renderings of the device in order to reveal the self-aligned and semi-vertical geometry. The 3D renderings in Fig. 1e were also focused on the device active region, so the electrodes were truncated to only show where TC and BC overlapped. To improve clarity, we have modified Fig. 1 in the following ways: (1) x-axis and y-axis labels are included in both Fig. 1d and Fig. 1e; (2) Fig. 1d and Fig. 1e now have separated (i), (ii)... labels; (3) Fig. 1e has been changed to better illustrate the alignment of TC and BC including arrows added to Fig. 1e (iii) to show where TC and BC would extend. In addition, we have updated the manuscript text to more thoroughly describe Fig. 1. Finally, we have added a figure to the supplementary information to complement Fig. 1e that shows the ALD etch mask in green.

2. Fig. 2a shows that rectifying diode has two polarities at VTG=6V and -6V. The authors attribute this behaviour to a possible band-to-band tunnelling. The two polarities behaviour was reported previously (Small 15, 1804661, 2019) by changing the back gate voltage. This is because MoS₂ and CNT work in different regime with different back gate voltage. I suggest the author to measure the transfer curve of MoS₂ and CNT, and the output curve of MoS₂ and CNT at different gate voltage, respectively. Furthermore, can the authors provide a detailed explanation based on the new results requested above?

Response:

Because polarity switching of the MoS₂-CNT system was reported initially in *Proc. Natl Acad. Sci. USA* **110**, 18076-18080 (2013) and then subsequently in *Nano Lett.* **18**, 1421-1427 (2018) and other papers including the one that the reviewer cites, we did not initially include an in-depth analysis of the individual semiconductor behaviors in this manuscript. However, for completeness, we now include new figures in the supplementary information that show the dual-gate transfer and output responses for the MoS₂ and CNT control transistors. We also added more text about the polarity switching in the main manuscript including the reference that the reviewer cited.

3. Fig. 2 and Fig. 3 shows that both top/back gates could modulate the heterojunction. So these two figures actually show the same information. I suggest the authors to combine Fig. 2 and Fig. 3 together and replace Fig. 3 with the dynamic response of the device with pulse measurement that is typically used to characterize the artificial neuron.

Response:

We apologize for not being more explicit concerning the important differences between Fig. 2 and Fig. 3. These figures are intentionally separated to highlight distinct operating modes for using the top and bottom gates to modulate the heterojunction. In Fig. 2 (independent gate operation), the bottom gate is set at a constant voltage throughout measurement. However, for Fig. 3 (dependent gate operation), the top and bottom gates are changed together throughout the measurement with a constant voltage offset. These two distinct operating modes have implications not only for the device electrostatic control (as demonstrated by Fig. 2 and Fig. 3) but also for the types of spiking behaviors they enable. For example, dependent gate operation enables Constant and Class I Spiking (Fig. 5a,b and Supplementary Fig. 20), whereas independent gate operation enables Latency, Integrator, and Phasic Spiking (Fig. 5c, d, e and Supplementary Fig. 21). We have updated the main manuscript text to better highlight the differences between Fig. 2 and Fig. 3.

4. In page 5, the authors state “The top gate can fully modulate the CNTs at all VBG, as evidenced by the negative transconductance (gm) in Fig. 2c. Due to low dielectric screening by CNT networks, the top gate can also modulate the n-type MoS₂ for VBG < 0 V, as evidenced by positive gm.” According to my understanding, it means the top gate can modulate the CNT as the drain current when VTG < 0 varies at all VBG. The top gate could also modulate the MoS₂ as the drain current when VTG > 0 also varies when VBG < 0. By claiming that the drain current shows dependence on gate voltage is clear enough, why do the authors mention gm without showing the gm curve? gm is typically used to show how good the gate could control the channel. Clearly, the authors just want to show that top gate could control both CNT and MoS₂. I suggest to rewrite these sentences.

Response:

The antiambipolar response of the heterojunction is most easily understood as the intersection of the n-type and p-type constituent transistor responses where there are two OFF states corresponding to when each transistor is OFF at large negative gate bias (n-type OFF) and large

positive gate bias (p-type OFF) and a peaked ON state corresponding to when both transistors are ON for intermediate gate biases. We have modified the main manuscript text to improve the clarity of this point. In addition, we have added new figures to the supplementary information to support the updated text: (1) Figure showing the dual-gate transfer response for MoS₂ and CNT control transistors; (2) Figure showing the transconductance, g_m , with respect to V_{TG} corresponding to Fig. 2c; (3) Added a figure panel showing the transconductance, g_m , with respect to V_{BG} to Supplementary Fig. 9.

5. In page 8, the authors state “This condition drives the GHeT through its peak current, I_{PEAK} , causing a sharp increase in the slope of V_m in proportion to $I_{syn} + I_{PEAK}$. As V_m continues to increase, $V_{TG} - V_m$ continues to decrease, accessing the positive g_m of the GHeT and allowing the delayed g_K channel to dominate and reset V_m . As V_m continues to increase, $V_{TG} - V_m$ continues to decrease, accessing the positive g_m of the GHeT and allowing the delayed g_K channel to dominate and reset V_m .” This sentence is very confusing and needs to be revised for better clarity.

Since the device already passed through its peak current, as V_m continues to increase, the $V_{TG} - V_m$ continues to decrease, so the g_m should go to the negative regime. Why the authors think it will access the positive g_m ? Moreover, as V_m continues to increase, the current will drop to 0 according to the I-V curve. Does the C2 start to charge T2 when the current drop to 0? What are the considerations when choosing C1 and C2?

Response:

When the circuit is first initialized by applying a constant I_{syn} , the GHeT is in an OFF state corresponding to a large positive gate bias (V_1). The capacitors then begin charging and V_m increases proportional to $I_{syn} + I_{OFF,GHeT}$. Once V_m reaches the threshold voltage of T1, T1 turns ON causing the voltage applied directly to the gates to drop quickly from V_1 to nearly 0 V. This quick drop in the voltage applied to the gates drives the GHeT device from its OFF state on the right side of the Gaussian response to the peak ON state, going through the region of negative transconductance. The dramatic increase in current from the GHeT, $I_{ON,GHeT}$, causes V_m to spike. As V_m increases the relative voltage applied to the gates ($V_{TG}-V_m$) continues to decrease, thus accessing the left side of the Gaussian response and resulting in a decreasing current and a positive transconductance. At this point, if the pull-down path, T2-R2-C2, corresponding to g_K was not present, V_m would continue to increase until the OFF state of the GHeT corresponding to a large negative gate bias was reached (as illustrated in Supplementary Fig. 13). However, with the pull-down path connected, V_m can only increase to the threshold of T2. As T2 turns ON, the pull-down path engages and V_m quickly drops below the threshold of T1 to reset the circuit. The gate of the pull-down transistor is connected to an RC delay path, therefore, the pull-down path self-terminates after the potential of V_m is sufficiently low. This sequence of events also configures the circuit to its reset state where synaptic current can again excite a spike at V_m by exploiting the GHeT response.

The values for C1 and C2 were determined by an initial approximation from simulations using raw data from a prototypical GHeT and then adjusted during prototyping to achieve optimal spiking. As the capacitance is decreased, the capacitors recharge from the constant I_{syn} and $I_{\text{OFF,GHeT}}$ more quickly, so the resting time between spikes decreases (i.e., increasing frequency of spiking) and then disappears altogether resulting in oscillations in V_m that do not mimic a biological neuron spike. For a small enough capacitance, the pull-down path is unable to drop V_m below the threshold of T1 and thus the circuit cannot reset. For increasing capacitance, the time increases for I_{syn} and $I_{\text{OFF,GHeT}}$ to charge up the capacitors to get V_m past the threshold of T1 and engage the circuit (i.e., decreasing frequency of spiking). We have updated the main manuscript text accordingly to better describe the operation of the spiking circuit. In addition, we have added the following content to the supplementary information: (1) Added a figure to show the spike evolution with respect to the GHeT; (2) Added a table to describe the state of various circuit elements at different points of the neuron spike; (3) Added a figure to show how the FWHM and spiking frequency change as a function of the capacitance and I_{syn} .

Below are some minor comments/questions about the manuscript:

1. There is a typo in the introduction. It should be “much” instead of “mush”.

Response: Thank you for pointing out the typo: “mush” was supposed to say “must” and has been corrected.

2. The authors claim wafer-scale production and a yield of 85%. Can the authors show the optical image of the device array fabricated over a 0.5 cm by 0.5 cm area?

Response:

The photolithography mask used to fabricate the GHeTs covers a 1 cm x 1 cm area, but the area of functioning devices is limited by the area of continuous monolayer MoS₂ grown via CVD, which covers ~ 0.5 cm x 0.5 cm. In the supplementary information, we have added a panel to Supplementary Fig. 12 that shows an optical image of the device array with a cm scale bar.

3. Figure 4c is very confusing. Please label source, drain, TG and BG of the GHeT clearly and explain the conductance vs time curve for g_K .

Response:

In the circuit, the source contact of the GHeT is connected to V_m , and the drain contact is connected to a set voltage. TG and BG are both connected to the delay. The conductance versus time curve for g_K comes directly from the Hodgkin-Huxley model for spiking neurons. We mimicked the Hodgkin-Huxley model behavior using the delayed turn-on of an NMOS transistor. The main text has been modified accordingly. In addition, labels were added to the GHeT structure in Fig. 4c to improve clarity.

Overall, this manuscript demonstrated the integrate-and-fire of an artificial neuron by integrating the Gaussian transistor into a circuit. However, the writing is very hard to read and the data analysis is unclear. Considering the broad readership of this journal with different background, I strongly suggest that the authors improve the manuscript to an acceptable level.

Response:

The feedback from all of the reviewers has guided the revision of our manuscript for better clarity and applicability to a broader audience.

Reviewer #2 (Remarks to the Author):

The authors present a new dual-gated device called Gaussian Heterojunction Transistor that shows very good tunability control, which allows (among possibly many other applications) its versatile use for building spiking neurons. I am not familiar with the fabrication processes nor device characteristics, so I cannot comment on that and compare with respect to alternative state-of-the-art solutions. From a system-level point of view of neuromorphic systems and applications, the solution proposed is certainly interesting. However, the authors should emphasize that more devices are required to build a spiking neuron, not just the new heterojunction transistor. For example, in Figs. 14-18 many other transistors plus passive elements (resistors and capacitors) are required.

Response:

It was not our intention to claim that the only circuit element required for our spiking neuron demonstration was the GHeT. However, the unique behavior of the GHeT is what enables the simplified spiking neuron circuits that we have demonstrated. We have updated the main manuscript text to more clearly delineate the additional circuit elements that are needed to achieve the reported spiking responses. Although additional elements are needed in our case, the resulting circuit is still considerably simpler and/or possesses greater versatility than alternative designs as we will be discussed further below.

The authors should compare their work to other alternatives. For example, <https://www.nature.com/articles/s41928-018-0023-2> shows a single device that can emulate a neuron just by adding a single capacitor or resistor. Authors should highlight their advantages with respect to this work.

Response:

The article that the reviewer cites presents an interesting approach to ANNs based on diffusive memristors as neurons and drift memristors as synapses. However, our current demonstration has additional advantages beyond circuit simplification that we believe make it a more versatile approach to neuron implementation. First, our circuit uses active devices that produce high gain swings to excite subsequent states of the neural network rather than passive components that require subsequent gain stages, which require more area and power for circuit implementation. Furthermore, the dual-gate tunability of the GHeT allows for runtime spiking threshold programmability as opposed to the spiking threshold being defined by the fabrication process. Notably, dynamic spiking threshold programmability is a key feature needed for both unsupervised learning (such as when implementing homeostasis (Carlson, Kristofor D., et al. Biologically plausible models of homeostasis and STDP: stability and learning in spiking neural networks. *The 2013 International Joint Conference on Neural Networks (IJCNN)*. IEEE, 2013)) as well as supervised learning (such as when implementing weight scaling by mapping a trained artificial neural network as SNN (Diehl, Peter U., et al. Fast-classifying, high-accuracy spiking deep

networks through weight and threshold balancing. *2015 International Joint Conference on Neural Networks (IJCNN)*. IEEE, 2015)). In a scalable system, dynamic spiking and threshold programmability will also be needed to correct process variability and process drift. Finally, instead of pursuing a neuron design that solely focuses on area reduction by minimizing the number of circuit elements, we present an approach that targets both low area and power operation as well as dynamic programmability of various spiking modes. In an $N \times N$ synaptic grid, area overhead for the synapses scale as $O(N^2)$ while the area overhead of neurons scale as $O(N)$. Therefore, while synapse design benefits substantially from a low area implementation, neuron design needs to balance other considerations such as interfacing with subsequent neural network states and runtime neural dynamic adaptation. We have updated the main manuscript text accordingly including the reference that the reviewer cited.

For fully CMOS solutions, they may compare against, for example, <https://ieeexplore.ieee.org/abstract/document/5537564>, or <https://ieeexplore.ieee.org/abstract/document/5372050>. These use a large number of transistors. But pioneering work of Cullurciello provided a very simple integrate-and-fire with a very reduced number of transistors, like <https://ieeexplore.ieee.org/abstract/document/1175509>.

Response:

In the first two articles that the reviewer cites (5537564 and 5372050), the CMOS demonstrations require ~ 20 transistors and individual voltage programming for each transistor in addition to other passive elements to achieve 6 different spiking modes. The third article that the reviewer cites (1175509) specifically targets a single spiking mode for image processing where spacing between spikes is related to the current level of a given pixel. In this demonstration, the event generator pixel (i.e., neuron) requires 7 transistors, a diode and a capacitor and is interfaced with a digital circuit containing 10 additional transistors.

In contrast, our work realizes 5 of the 8 spiking modes with one GHeT, two transistors, two capacitors, and two resistors. The additional 3 spiking modes require additional elements but even our most complex circuit (Supplementary Figure 24) has fewer elements compared to the aforementioned CMOS demonstrations. We have updated the main manuscript text accordingly including the references that the reviewer cited.

Reviewer #3 (Remarks to the Author):

In this manuscript (entitled “Spiking Neurons from Tunable Gaussian Heterojunction Transistors”), the authors proposed spiking neurons based on dual-gated tunable Gaussian heterojunction transistors with MoS₂ and CNTs. Hodgkin-Huxley spiking neuron was realized using such a Gaussian heterojunction transistor. With simulations, various biological spiking responses including phasic spiking, delayed spiking, and tonic bursting could be obtained. In general, this work is interesting and suitable for Nature Communications. There are some comments to the authors:

(1) What is the energy consumption per spike of the device? Please compare it with the energy consumption with the biological neurons and artificial neurons developed by other types of devices, metal-insulator-transition (MIT) devices, for example.

Response:

The energy consumption of the GHeT-based spiking neuron circuit is ~250 nJ/spike with a FWHM of ~200 ms. Since this is the first demonstration of our dual-gate tunable GHeT and the first demonstration of GHeT-based spiking neuron circuits, various aspects of the device and components in the circuit have not yet been optimized for the lowest energy consumption. However, with device scaling and on-chip integration, the energy consumption of GHeT-based spiking neuron circuits would easily be comparable to that of MIT-based spiking neurons (~200 pJ/spike, *Front. Neurosci.* **12**, 210 (2018)) and likely approach the energy consumption levels of the most simplified CMOS-based biomimetic spiking neurons at comparably scaled dimensions (~78 fJ/spike, *Front. Neurosci.* **11**, 123 (2017)). Scaling modifications to the GHeT include decreasing the gate dielectric thicknesses and reducing the device channel width. Decreasing the gate dielectric thicknesses will lower the gate operating voltage while also improving electrostatic control of the device OFF states. In particular, reduction from 4 V (current demonstration) to 1 V gate operating voltage would decrease energy consumption by at least an order of magnitude. Since the capacitances in the circuit are correlated to the magnitude of $I_{\text{syn}} + I_{\text{OFF}}$, improved electrostatic control of the OFF states of the GHeT would also allow the capacitances to be decreased substantially, which in turn would decrease the FWHM of the neuron spikes. Since the energy consumption is proportional to charging/discharging capacitance, decreasing the capacitors from a few hundred nF (current demonstration) to a few hundred fF (as expected in an integrated circuit) would drop energy consumption by five to six orders of magnitude. Furthermore, on-chip integration and custom design of transistors T1 and T2 would also decrease energy consumption. We have updated the main manuscript text accordingly.

(2) Please explain why MoS₂ and CNTs are selected? And compare it with other materials such as oxide semiconductors.

Response:

The material selection considerations specific to the device geometry included: (1) atomically thin materials for strong electrostatic modulation; (2) flexibility of at least one of the semiconducting materials to conform over the various steps in the device structure; (3) compatibility with large-area photolithography. The additional considerations for electronic behavior were: (4) materials that would form a p-n junction based on carrier type and band alignment; (5) materials that would exhibit antiambipolar behavior based on suitable threshold voltages (i.e., $V_{th,p\text{type}} > V_{th,n\text{type}}$). We selected MoS₂ as the n-type semiconductor because it can be grown as continuous monolayer films via chemical vapor deposition. We selected solution-processed CNTs as the second semiconductor due to its compatibility with large-area processing and arbitrary surface topographies, controllable p-type/ambipolar behavior, and desirable threshold voltage and band alignment with MoS₂.

Note that large-area lateral heterojunctions from semiconducting oxides (e.g., IGZO) and solution-processed CNTs have also shown desired antiambipolar behavior (*Nano Lett.* **15**, 416-421 (2015)), so we did attempt to use oxide semiconductors in some of our self-aligned devices. However, we found incompatibilities with our specific device structure. For example, because the mobility of the semiconducting oxides was lower than that of MoS₂ or CNTs, the drain and gate voltages that were necessary to get useful current levels were beyond the limits of our thin, self-aligned dielectrics, resulting in poor device yield due to dielectric breakdown and leakage. Additionally, to achieve full coverage of the device structure, the oxide films had to be thicker (i.e., not atomically thin like MoS₂ or CNTs) and thus were not sufficiently gate tunable. We have updated the main manuscript text accordingly. In addition, we added the dual-gate transfer and output responses for MoS₂ and CNT control transistors.

(3) As the heterojunction transistors can be tuned by VTG and VBG, can the properties of the spiking neurons, such as spiking frequency, based on the heterojunction transistors be modulated by VTG and VBG? If yes, please show the modulated properties.

Response:

Both V_{TG} and V_{BG} are used in all the spiking neuron circuits that we presented and are manipulated to enable the variety of different spiking behaviors presented in Fig. 5. In Supplementary Fig. 18 for experiment and Supplementary Fig. 19 for simulations, we show that the voltage offset between the two gates for dependent biasing can be used to modulate the spiking response from no spiking to constant spiking with increasing spiking frequency. In Fig. 5, we show that dependent biasing of the gates enables Constant and Class I Spiking (circuit in Supplementary Fig. 20) and independent biasing of the gates enables Latency, Integrator, and Phasic Spiking (circuit in Supplementary Fig. 21). Additional modifications to the dual-gate biasing enables Phasic Bursting (circuit in Supplementary Fig. 22), Tonic Bursting (circuit in Supplementary Fig. 23), and Dampened Tonic Bursting (circuit in Supplementary Fig. 24). We have updated the main manuscript text accordingly.

Reviewers' comments:

Reviewer #1 (Remarks to the Author):

The authors have carefully revised the manuscript, and I am satisfied with the revision made. Considering the importance of this work to neuromorphic computing, I recommend the publication of this manuscript in Nature Communications.

Reviewer #2 (Remarks to the Author):

I am happy with the modified version. The authors addressed all my comments satisfactorily.

Reviewer #3 (Remarks to the Author):

The manuscript was carefully revised based on the referee's comments, and now it can be accepted for publication after minor revision. Here are some comments for further revision.

1) Authors reported "While memristors¹¹, memtransistors^{12,13}, domain-wall memories¹⁴, metal-insulator-transition (MIT) devices¹⁵, and Gaussian synapses¹⁶ have been developed.....".in the second paragraph. In order to give more detailed background knowledge to the reader, some references on the topic of dual-gate/multi-gate transistors for neuromorphic application should be cited. For example:

1) 2D MoS₂ Neuromorphic Devices for Brain-Like Computational Systems. Small. 13, 1700933, (2017);

2) Multi-Gate Synergic Modulation in Laterally Coupled Synaptic Transistors. Applied Physics Letters. 107, 143502 (2015).

2) As we know, the operation frequency of the biological neurons is several tens Hz. The operation frequency of the reported spiking neurons based on gaussian heterojunction transistors is very low (less than 1.0 Hz). Is there any method to increase the operation frequency of the proposed neurons to several tens Hz?

Reviewer #1 (Remarks to the Author):

The authors have carefully revised the manuscript, and I am satisfied with the revision made. Considering the importance of this work to neuromorphic computing, I recommend the publication of this manuscript in Nature Communications.

Reviewer #2 (Remarks to the Author):

I am happy with the modified version. The authors addressed all my comments satisfactorily.

Reviewer #3 (Remarks to the Author):

The manuscript was carefully revised based on the referee's comments, and now it can be accepted for publication after minor revision. Here are some comments for further revision.

1) Authors reported “While memristors¹¹, memtransistors^{12,13}, domain-wall memories¹⁴, metal-insulator-transition (MIT) devices¹⁵, and Gaussian synapses¹⁶ have been developed.....” in the second paragraph. In order to give more detailed background knowledge to the reader, some references on the topic of dual-gate/multi-gate transistors for neuromorphic application should be cited. For example:

- 1) 2D MoS₂ Neuromorphic Devices for Brain-Like Computational Systems. *Small*. 13, 1700933, (2017);
- 2) Multi-Gate Synergic Modulation in Laterally Coupled Synaptic Transistors. *Applied Physics Letters*. 107, 143502 (2015).

Response:

We have updated the main manuscript text to include the references that the reviewer cited.

2) As we know, the operation frequency of the biological neurons is several tens Hz. The operation frequency of the reported spiking neurons based on gaussian heterojunction transistors is very low (less than 1.0 Hz). Is there any method to increase the operation frequency of the proposed neurons to several tens Hz?

Response:

Many of the strategies for scaling down energy consumption would also increase the operating frequency to the desired range for biological neurons. For example, decreasing the gate dielectric thicknesses and reducing the device channel width would lower operating voltages and improve electrostatic control of the device OFF state. Since the capacitances in the circuit are correlated with the magnitude of $I_{\text{syn}} + I_{\text{OFF}}$, improved electrostatic control of the OFF states of the GHeT will also allow the capacitances to be decreased substantially. We have updated Supplementary Figure 17a to include simulated frequency and FWHM of the spiking response for capacitances down to 0.1 nF. For capacitance values on the order of nF, the spiking frequency is in the range of tens of Hz, which matches the range for biological neurons. The manuscript and supplementary information have been updated accordingly.

REVIEWERS' COMMENTS:

Reviewer #3 (Remarks to the Author):

The authors have carefully revised the manuscript based on the comments, and I recommend the publication of this revised manuscript in Nature Communications.